# Prognostic Value of Lectin-like Oxidized Low-Density Lipoprotein Receptor-1 for Future Cardiovascular Disease Risk and Outcome: A Systematic Review and Meta-Analysis

**DOI:** 10.3390/biomedicines13020444

**Published:** 2025-02-12

**Authors:** Amilia Aminuddin, Nazirah Samah, Nur Aishah Che Roos, Shawal Faizal Mohamad, Boon Cong Beh, Adila A. Hamid, Azizah Ugusman

**Affiliations:** 1Department of Physiology, Faculty of Medicine, Universiti Kebangsaan Malaysia, Jalan Yaacob Latif, Bandar Tun Razak, Cheras, Kuala Lumpur 56000, Malaysia; p124809@siswa.ukm.edu.my (N.S.); adilahamid@ppukm.ukm.edu.my (A.A.H.); 2Faculty of Medicine and Defence Health, National Defence University of Malaysia, Kem Sungai Besi, Kuala Lumpur 57000, Malaysia; nuraishah@upnm.edu.my; 3Department of Cardiology, Universiti Kebangsaan Malaysia, Jalan Yaacob Latif, Bandar Tun Razak, Cheras, Kuala Lumpur 56000, Malaysia; drshawal81.hctm@ukm.edu.my (S.F.M.); dr.behbooncong@ppukm.ukm.edu.my (B.C.B.)

**Keywords:** cardiovascular disease, coronary artery disease, acute coronary syndrome, myocardial infarction, stroke, soluble lectin-like oxidized LDL receptor-1, risk

## Abstract

Cardiovascular disease (CVD) remains a leading cause of mortality globally, underscoring the need for robust predictive biomarkers to enhance risk stratification. Soluble lectin-like oxidized low-density lipoprotein receptor-1 (sLOX-1) has emerged as a promising biomarker linked to oxidative stress and endothelial dysfunction, both critical mechanisms in atherogenesis and cardiovascular events. **Objectives:** This study aimed to evaluate the prognostic value of sLOX-1 in predicting major adverse cardiovascular and cerebrovascular events (MACCEs), myocardial infarction (MI), heart failure (HF), and stroke outcomes through a systematic review and meta-analysis. **Methods:** A systematic literature search was conducted across PubMed, Scopus, Web of Science, and Ovid databases for studies published between 2014 and October 2024. Eligible studies assessed the association between sLOX-1 levels and future CVD outcomes in adult populations. Meta-analysis pooled hazard ratios (HRs) were assessed using random- and fixed-effects models. Heterogeneity was evaluated using the *I*^2^ statistic, and study quality was assessed using the Newcastle–Ottawa Scale. **Results:** Fourteen studies were included, encompassing diverse populations with coronary artery disease (CAD), acute coronary syndrome (ACS), or stroke, with follow-up durations ranging from 30 days to 19.5 years. The meta-analysis of three studies on CAD patients demonstrated a significant association between elevated sLOX-1 levels and increased MACCE risk (HR: 2.3, 95% CI: 0.99–5.33, *p* = 0.05), albeit with high heterogeneity (*I*^2^ = 83%). The fixed-effects analysis yielded a more consistent HR of 1.47 (95% CI: 1.19–1.81, *p* < 0.01). **Conclusions:** sLOX-1 shows promising potential as a prognostic biomarker for CVD and is associated with an increased risk of MACCEs in CAD patients. However, the high heterogeneity among the included studies highlights the need for standardized protocols and larger, well-designed prospective studies to validate its clinical utility. The integration of sLOX-1 into risk prediction models could improve CVD management by identifying high-risk individuals for targeted interventions.

## 1. Introduction

Cardiovascular disease (CVD) remains the leading cause of morbidity and mortality worldwide, accounting for an estimated 17.9 million deaths each year, representing 31% of all global deaths [1]. CVD encompasses a broad spectrum of heart and vascular disorders, including coronary artery disease (CAD), stroke, and peripheral artery disease. In Malaysia, the burden of CVD is alarming, with ischemic heart disease identified as the leading cause of death, contributing to 17% of total fatalities in 2020 [2]. The rising prevalence of risk factors such as hypertension, diabetes, obesity, and smoking has further escalated the incidence of CVD in the country [3]. Given the substantial health and economic toll of CVD, enhancing early detection and prevention strategies is imperative. Identifying reliable biomarkers to predict future cardiovascular events is vital for mitigating the impact of CVD on individuals and healthcare systems.

In recent years, the search for reliable biomarkers has become an integral focus of cardiovascular research, aimed at improving the early diagnosis and prognosis of CVD. Biomarkers provide insight into underlying pathophysiological processes and can help identify individuals at high risk for cardiovascular events, even before clinical symptoms manifest [4]. Traditional markers, such as high-sensitivity C-reactive protein (hs-CRP) and troponins, have been widely used, yet their predictive value may be limited in certain populations [5]. This has led to the investigation of novel biomarkers with greater specificity and accuracy. One such promising marker is soluble lectin-like oxidized low-density lipoprotein receptor-1 (sLOX-1), a receptor involved in the binding and uptake of oxidized LDL (oxLDL), which plays a pivotal role in the development and progression of atherosclerosis [6]. Elevated levels of sLOX-1 have been associated with endothelial dysfunction, plaque instability, and an increased risk of acute coronary events, making it a potential candidate for improving CVD risk stratification [7].

sLOX-1 is a truncated, circulating form of the membrane-bound LOX-1 receptor, which is predominantly expressed on endothelial cells, macrophages, and smooth muscle cells within atherosclerotic plaques [8]. Structurally, LOX-1 belongs to the C-type lectin family and consists of four main domains: a short N-terminal cytoplasmic domain, a transmembrane domain, a neck region, and an extracellular C-type lectin-like domain [9]. The extracellular domain is responsible for binding oxLDL, a key driver of atherogenesis. Under conditions of oxidative stress and inflammation, oxLDL accumulates in the vascular endothelium, triggering LOX-1 activation [10]. This binding promotes endothelial dysfunction, foam cell formation, and plaque instability, all of which contribute to atherosclerosis progression.

The soluble form, sLOX-1, is released into circulation upon cleavage of the membrane-bound receptor, with elevated serum levels observed in patients with acute coronary syndrome (ACS) and other cardiovascular disorders. These findings suggest its potential as a non-invasive biomarker for CVD [11,12,13]. sLOX-1 has been linked to endothelial dysfunction, a critical early event in the development of atherosclerosis, driven by factors such as hypertension, diabetes, and hyperlipidemia [14]. Elevated sLOX-1 levels have also been associated with increased inflammatory markers such as interleukin-6 (IL-6) and tumor necrosis factor-alpha (TNF-α), both of which exacerbate atherogenic processes [15]. Clinical investigations have demonstrated a correlation between sLOX-1 levels and the severity of atherosclerosis in patients with suspected CAD [16], reinforcing its relevance as a predictive biomarker in CVD management.

Beyond its association with acute events, growing evidence suggests that sLOX-1 could be a valuable predictor of future CVD events. Several longitudinal studies have investigated the relationship between baseline sLOX-1 levels and the incidence of cardiovascular outcomes such as myocardial infarction (MI), heart failure (HF), and stroke. For instance, baseline sLOX-1 concentrations have been shown to predict MI and heart failure in general populations [17]. Among patients with established CAD, higher sLOX-1 levels have been linked to increased long-term mortality and recurrent ischemic events [18]. However, despite the accumulating data, the overall consistency and strength of these associations remain unclear. To address this, a systematic review and meta-analysis were conducted to quantitatively assess the predictive value of sLOX-1 for future CVD events. This meta-analysis aimed to provide a comprehensive and reliable estimate of the association, explore potential sources of heterogeneity across studies, and further clarify sLOX-1′s role in cardiovascular risk stratification.

## 2. Methodology

This review was conducted in accordance with the Preferred Reporting Items for Systematic Reviews and Meta-Analysis (PRISMA) guidelines [19]. The PRISMA checklist is reported in Appendix A (Table A1 and Table A2). The review protocol has been registered with the International Platform of Registered Systematic Review Protocols (registration number INPLASY2024120078) [20].

### 2.1. Research Question and Search Strategy

The research question was formulated using the “PICO” framework. Patients with ACS, CAD, and acute ischemic stroke (AIS) or healthy subjects were identified as the “Population (P)”. Low sLOX-1 levels were defined as the “Comparison (C)”, while the occurrence of major adverse cardiocerebrovascular events (MACCEs), MI, HF, recurrent stroke, or unfavorable outcomes post stroke constituted the “Outcome (O)”. sLOX-1 levels were designated as the “Intervention (I)”. Hence, the research question was as follows: are increased sLOX-1 levels predictive of future CVD? A comprehensive literature search was carried out across four databases (Ovid, Scopus, Pubmed, and Google Scholar), focusing on studies published between 2014 and October 2024 (Accessed on 1 November 2024). The search employed the following keywords: (“LOX-1” OR “lectin-like oxidized LDL receptor-1”) AND (“Cardiovascular disease” OR “CVD” OR “Coronary artery disease” OR “CAD” OR “Myocardial infarction” OR “Acute coronary syndrome” OR “Atherosclerosis” OR “Peripheral vascular disease” OR “Cerebrovascular accident” OR “Stroke” OR “Major cardiovascular event” OR “MACE” OR “Mortality”).

### 2.2. Study Criteria

Two researchers (NS and AA) independently reviewed each article, ensuring compliance with predefined inclusion and exclusion criteria. The inclusion criteria included the following: (1) full-text peer-reviewed original articles published in English, (2) prospective and retrospective studies investigating the relationship between sLOX-1 and future coronary artery disease (CAD), cardiovascular disease (CVD), or major adverse cardiovascular events (MACEs), and (3) studies involving adult patients diagnosed with ACS, CAD, or AIS or healthy subjects, regardless of gender. The exclusion criteria consisted of (1) articles not published in English, (2) reviews, conference abstracts, editorials, newsletters, books, and book chapters, (3) in vitro studies, and (4) research involving animals.

### 2.3. Article Selection and Data Extraction

The article selection process comprised three stages. First, articles were screened according to their titles and types, with review or editorial articles excluded. Next, abstracts were examined to remove any articles that were irrelevant to sLOX-1 and CAD, CVD, or MACEs. Finally, the remaining articles underwent a detailed full-text review, and those that did not meet the inclusion criteria were eliminated. The article selection process was conducted independently by two reviewers (A.A. and N.S.). Similarly, data extraction was performed independently by two reviewers (A.A. and N.S.) using a standardized Excel form. Data extracted included study design, study duration, participant characteristics (e.g., age, gender, CVD diagnosis), methods of sLOX-1 measurement, and the relationship between sLOX-1 and atherosclerosis. Any disagreement was resolved by consensus or consultation with a third reviewer (A.U.) if necessary.

### 2.4. Risk of Bias Assessment

Two reviewers (N.S. and A.A.) independently assessed the risk of bias in the selected articles using the Newcastle–Ottawa Scale (NOS) [21]. For cohort and cross-sectional studies, the NOS evaluated the selection of study groups (exposed vs. non-exposed), their comparability, and outcome assessment. In contrast, for case–control studies, the NOS evaluated the selection of study groups (cases vs. controls), their comparability, and exposure assessment. Each of the eight NOS items was rated with one or two stars. Studies that received a total score of seven to nine stars were classified as high-quality, those with four to six stars as fair-quality, and those with one to three stars as low-quality.

### 2.5. Statistical Analysis

A meta-analysis was conducted using Review Manager (RevMan version 5.4 software, The Cochrane Collaboration 2000, Denmark) [22]. The hazard ratio (HR) or odds ratio (OR), with a 95% confidence interval (CI), derived from multivariate Cox proportional hazard analysis, were used as the effect estimate to evaluate s-LOX-1 as a predictor of MACEs in patients with ACS or AIS. The heterogeneity among studies was assessed using (1) the chi-squared test, with a *p*-value of less than 0.10 indicating statistical significance, and (2) the Higgin’s *I*^2^ statistic [23]. An *I*^2^ value of ˂25% indicated low heterogeneity, while an *I*^2^ value of ≥75% indicated high heterogeneity. Due to the limited number of studies available for meta-analysis and varying study population across studies, the random-effects (RE) model was employed. Statistical significance was set at *p* ˂ 0.05. Subgroup analysis was not performed due to the limited number of studies. In addition, funnel plots or the Egger test for publication bias were not conducted, as fewer than 10 studies were included.

## 3. Results

The search retrieved a total of 9918 articles from four databases: Ovid (1490), Scopus (7925), Pubmed (500), and Google Scholar (3). Articles were compiled using Mendeley Desktop Version 1.19.8 (Mendeley Ltd., London, UK) and 821 duplicates were removed. Additionally, 4155 reviews, editorials, and non-English-language articles were excluded. Title and abstract screening excluded 4908 articles unrelated to sLOX-1, CAD, or MACEs. The remaining 34 articles were read in full, with only 14 articles meeting the eligibility criteria for inclusion. The study selection process is summarized in Figure 1. The quality assessment of the 14 selected articles using the NOS scale is outlined in Table 1 and Table 2. The scores ranged from 6 to 8, reflecting fair to high quality. Specifically, 12 studies were categorized as high-quality [13,17,18,24,25,26,27,28,29,30,31,32], while 2 studies were classified as fair-quality [33,34]. Altogether, the risk of bias for all included studies in terms of the selection, comparator, and outcome reporting bias was considered low. Therefore, results synthesized using data from the included studies may contribute to a reliable source of information which could be used to inform clinical practice regarding the association between sLOX-1 and future CVD risk.

### 3.1. General Characteristics of the Included Studies

The 14 studies included in this study were published between 2014 and October 2024. Seven studies focused on the predictive value of sLOX-1 for CAD, while another seven focused on the role of sLOX-1 as predictor for stroke. All studies were prospective, except for the study by Ren et al. 2023 [28], which was retrospective. Two studies involved subjects without CAD or stroke at baseline [17,32], while the remaining studies involved subjects who already had CAD or ACS [18,24,25,26,27,33] or stroke [13,19,28,29,30,31]. The duration of follow-up ranged from 30 days to 19.5 years. Most of the subjects were male, with ages ranging from middle-aged to elderly. All studies assessed sLOX-1 levels from plasma or serum samples. For CAD or ACS patients, outcome measures included MACCEs, HF, or MI. For stroke patients, outcomes focused on functional recovery post stroke and stroke recurrence. The definition of MACCEs varied among studies. For instance, Zhao (a) et al. 2019 [33] defined MACCEs as all-cause death, readmission for ACS, unplanned repeat revascularization, definite stent thrombosis, and ischemic stroke, while Zhao (b) et al. 2019 [26] defined them as all-cause death, nonfatal acute MI, and readmission for Braunwald’s class IIIb unstable angina requiring treatment. In addition, Higuma et al. 2015 [27] defined MACCEs as cardiovascular mortality and recurrent nonfatal MI. Several studies used different cut-off values to define high sLOX-1 levels. For instance, Higuma et al. defined it as >71 pg/mL, while Zhao et al. [33] used ≥1.10 ng/mL, and Zhao et al. [26] applied >0.91 ng/mL. For sLOX-1 measurements, the ELISA technique was used in most of the studies [13,18,24,25,26,29,30,34]. However, there were other methods used, such as the PEA technique [17], sandwich chemiluminescent enzyme immunoassay [27], anti-analyzer [31], and multiplex kit (34).

### 3.2. Summary of Findings

#### 3.2.1. sLOX-1 and Future MACCEs, MI and HF

For studies investigating the association between sLOX-1 and future MACCEs, MI, or HF, most subjects had ACS or CAD, with only one study involving a non-CAD population [17]. Among CAD or ACS patients, five studies reported a significant positive association (Table 3). For example, Higuma et al. 2015 [27] found that higher sLOX-1 levels predicted a higher risk of MACCEs, with a HR of 2.44 (95% CI: 1.07–5.57), while Zhao (a) and (b) et al. 2019 [26,33] reported HRs of 1.28 (95% CI: 1.02–1.60) and 4.73 (95% CI: 2.17–10.30), respectively. The authors of [24] also reported that elevated sLOX-1 levels were associated with a 2.3-fold increased risk of cardiovascular mortality within one year of follow-up (HR: 2.29, 95% CI: 1.19–5.34; *p* = 0.0148). In another study, sLOX-1 levels were associated with MACCE in ST-elevation MI (STEMI) and unstable angina (UA)/non-STEMI groups (Spearman correlation coefficients: 0.345, *p* < 0.001, and 0.189, *p* = 0.017, respectively) [18]. Among patients with recurrent cases of MI and stroke, the baseline post interventional sLOX-1 levels were higher compared to the non-recurrent cases [25]. In a study of non-CAD subjects, higher sLOX-1 levels were associated with an increased risk of MACCEs (HR: 1.76, 95% CI: 1.40–2.21) [17].

#### 3.2.2. sLOX-1 and Future Functional Outcomes Post Strokes or Recurrent Strokes

Among stroke patients, several studies identified significant associations between baseline sLOX-1 and unfavorable or favorable functional outcomes post strokes [13,29,30,34] (Table 4). Functional outcomes were measured using the Modified Rankin Score (mRS). Additionally, several studies reported a significant association between sLOX-1 levels and higher risk of recurrent stroke among patients with or without prior history of stroke [28,31,32].

#### 3.2.3. Meta-Analysis

##### sLOX-1 and MACCEs

Our meta-analysis of three studies [26,27,33] examined the prognostic value of sLOX-1 for predicting MACCE in CAD patients. Using a random-effects model, it was found that higher sLOX-1 levels were associated with a 2.3-fold increased risk of MACCE (HR: 2.3, 95% CI: 0.99–5.33, *p* = 0.05) (Figure 2). However, the wide confidence interval and high heterogeneity (*I*^2^ = 83%) indicated variability across the studies. To address heterogeneity, a sensitivity analysis using a fixed-effects model was performed (Figure 3). This analysis yielded a more precise HR of 1.47 (95% CI: 1.19–1.81, *p* < 0.01), confirming a consistent and statistically significant association between elevated sLOX-1 levels and increased MACCE risk. There were several potential sources of variability. First, there were differences in study population, such as age and prevalence of comorbidities. Although the mean age of the subjects in the three studies was relatively similar (67–69 years old), the age range were different. For example, one study by Zhao et al. [26] involved subjects with age ranges between 32 and 87 years old, while the other study by Zhao et al. [33] involved subjects with age ranges between 59 and 76 years old. In addition, each study had different distributions of comorbidities. For example, in the study population in Higuma et al. [27], 75% had hypertension, 33% had diabetes, 63% had dyslipidemia, and 54% smoked. In Zhao et al. [33], among those with MACCEs, 50% smoked, 31.3% had diabetes, and 58.26% had hypertension. In the study population in Zhao et al. [26], 72% had hypertension, 45% smoked, and 45% had diabetes. Variations in comorbidities also contributed to variations in the medications used. Most of the populations in the three studies used statins, angiotensin-converting enzyme inhibitors, angiotensin II receptor blockers, and beta blockers; however, the percentages that used them were different [26,27,33]. Another possible contributing factor was the method used for sLOX-1 measurements. The studies by Zhao et al. ([26,33]) used ELISA, while the study by Higuma et al. [27] used a sandwich chemiluminescent enzyme immunoassay. Moreover, the definition of outcome was also differed across the studies, as mentioned above.

##### sLOX-1 and Stroke Outcomes

A meta-analysis for sLOX-1 and stroke outcomes could not be conducted due to high variation between the studies. Although there were three studies that investigated the association between baseline s-LOX-1 and functional outcomes post stroke [13,29,34], the patients included were not similar. For example, the study by Yan et al. [29] involved patients with first-ever stroke, while Zheng et al. [34] involved patients with first-ever stroke and recurrent ischemic stroke. In another study, this characteristic was not detailed [13]. In addition, there were differences in patient age, comorbidities, the medications used, and the duration of follow-up between studies. For example, the studies by Zheng et al. [34] and Yan et al. [29] assessed the patient condition at three months, while the study by Li et al. [13] reviewed the patient condition at one year. Because of these variation, the pooling of data was deemed inappropriate.

## 4. Discussion

Our meta-analyses involving CAD patients supported the role of sLOX-1 as a prognostic biomarker for MACCE. The stronger association observed in the fixed-effects model suggests that sLOX-1 may have consistent predictive value when study-level variations are accounted for. However, the high heterogeneity observed in both models necessitates caution when interpreting the pooled estimates. Possible contributors to this heterogeneity include differences in study populations (e.g., age, comorbidities, and treatment regimens), sLOX-1 measurement methods and threshold definitions, MACCE definitions, and follow-up durations. The variability in sLOX-1 cut-off values reported across studies may also reflect underlying differences in systemic inflammation, as inflammation is a known contributor to elevated sLOX-1 levels. However, most of the studies included in the meta-analysis did not report data on CRP, limiting our ability to normalize sLOX-1 levels based on inflammation. Despite this limitation, the robust association between sLOX-1 levels and cardiovascular outcomes observed in most of the studies suggests that sLOX-1 remains a valuable biomarker, even without adjustment for inflammatory status.

The integration of sLOX-1 into clinical practice offers several potential pathways to enhance cardiovascular care. As a biomarker associated with oxidative stress and endothelial dysfunction, sLOX-1 could serve as an adjunct to traditional risk scores, such as the Framingham Risk Score or GRACE score, to improve risk stratification, particularly for individuals at intermediate or high risk. Additionally, its role in the early detection of subclinical atherosclerosis or acute vascular events could be pivotal in identifying individuals with ambiguous symptoms or those requiring timely intervention. sLOX-1 levels may also inform personalized treatment decisions, such as guiding the use of statins, antioxidants, or anti-inflammatory therapies, and monitoring these levels could help assess treatment efficacy over time. Furthermore, in acute coronary syndrome patients, sLOX-1 could refine prognostic models, aiding in decisions regarding invasive procedures or extended monitoring during hospital admission and follow-up. Lastly, its utility in specific high-risk populations, such as individuals with diabetes, metabolic syndrome, or chronic inflammatory conditions, could enable targeted screening and earlier intervention to mitigate cardiovascular risk. However, integrating sLOX-1 into routine clinical practice faces several challenges. Firstly, the variability in sLOX-1 measurement techniques, including the use of different ELISA kits and assay methodologies, poses a significant challenge. The use of diverse kits, each employing antibodies with varying specificities and affinities, can substantially influence protein measurement [35]. Additionally, differences in detection limits and quantification thresholds among kits may lead to the underestimation of low protein concentrations when low-sensitivity kits are used [36]. These inconsistencies complicate result comparability and the establishment of universal reference ranges. Furthermore, there is no consensus on clinically meaningful sLOX-1 cut-off values for risk stratification. Many studies report associations between sLOX-1 levels and outcomes but fail to define threshold categories, such as low, moderate, or high risk [17]. Lastly, integrating sLOX-1 testing into routine clinical workflows requires the development of cost-effective, scalable methods and collaborative efforts between researchers and clinicians to establish a standardized interpretation of sLOX-1 levels for informed clinical decision-making.

Several mechanisms have been implicated in the association between sLOX-1, MACCEs, and post-stroke functional outcomes. First, sLOX-1 is associated with oxidative stress and endothelial dysfunction. Its ligand, oxLDL, disrupts eNOS activity in endothelial cells, reducing nitric oxide (NO) production. Lower NO levels lead to endothelial dysfunction, promoting damage, lipid buildup, and plaque formation [37]. Elevated sLOX-1 levels reflect oxidative stress, impair endothelial function, and contribute to atherosclerosis, plaque rupture, and thrombus formation [10]. High circulating sLOX-1 levels are linked to an increased likelihood of thrombotic events and worse outcomes in acute cardiovascular conditions [8]. Second, sLOX-1 release is associated with the activation of pro-inflammatory pathways, leading to the upregulation of adhesion molecules. Activation of mitogen-activated protein kinase (MAPK) by oxLDL promotes the transcription of adhesion molecules, including monocyte chemoattractant protein-1 (MCP-1), intracellular adhesion molecules-1 (ICAM-1), and vascular cell adhesion molecules (VCAM-1). Elevated MCP-1 attracts monocytes to the endothelium [38,39,40]. This inflammatory response exacerbates atherosclerosis and predisposes individuals to adverse cardiovascular and cerebrovascular events, impeding poor functional recovery after a stroke. Third, in atherosclerosis, sLOX-1 contributes to vascular remodeling, promoting the migration and proliferation of smooth muscle cells and the production of matrix metalloproteinases (MMPs) [41,42]. The binding of oxLDL to LOX-1 on endothelial cells activates the protein kinase C-beta (PKC-β) signaling cascade. PKC-β increases the expression of MMPs [41]. Excessive MMP activity degrades the extracellular matrix and contributes to endothelial dysfunction, vascular remodeling, and plaque instability. This process increases the vulnerability of plaques to rupture, thereby increasing the risk of acute cardiovascular events, including MI and stroke [41,42]. In stroke, elevated sLOX-1 levels are linked to increased vascular permeability and damage, correlating with poorer functional outcomes and recovery [43]. The detailed mechanisms underlying the role sLOX-1 in CVD pathophysiology have been extensively reviewed in previous studies [10,44].

## 5. Limitations and Future Directions

The exclusion of gray literature and unpublished data is a limitation of this review. While this decision was made to prioritize the quality and rigor of the evidence, it may have resulted in the omission of relevant studies, potentially introducing publication bias. In addition, a restriction of this meta-analysis lies in the limited number of available studies, which restricts the statistical power and the robustness of the pooled estimates. A small number of studies inherently increases the influence of individual study results on the overall meta-analysis, potentially biasing the conclusions. Furthermore, the wide confidence intervals observed in the random-effects model highlight imprecisions, necessitating caution in interpretation.

High heterogeneity across studies also raises concerns about the consistency and comparability of results. Variations in study design, population characteristics, sLOX-1 assay methodologies, and follow-up durations challenge the generalizability of the findings. Standardized protocols in future research are needed to address this issue. Publication bias and potential unmeasured confounders within the included studies cannot be excluded, further limiting the reliability of the results. Well-designed prospective studies are urgently needed to confirm the prognostic value of sLOX-1, particularly for MACCEs and functional outcomes post stroke. Such studies should recruit diverse populations to enhance the generalizability of findings across demographic and clinical subgroups. Efforts should also focus on standardizing sLOX-1 measurement protocols, including assay methods, sampling techniques, and risk stratification thresholds. A consensus on these parameters will facilitate the clinical integration of sLOX-1 and allow for meaningful comparisons between studies.

Additionally, further investigations into the biological mechanisms linking sLOX-1 and MACE are critical. Understanding sLOX-1′s role in endothelial dysfunction, inflammation, and plaque instability could validate its clinical relevance and uncover novel therapeutic targets for CVD. Exploring these mechanisms may also clarify sLOX-1′s potential utility in guiding treatment strategies or monitoring disease progression.

## 6. Conclusions

This systematic review highlights the potential of sLOX-1 as a prognostic biomarker for CVD, particularly in predicting MACCEs. Despite promising findings, limitations such as the small number of studies, wide confidence intervals, and significant heterogeneity across studies underscore the need for cautious interpretation. While sLOX-1 shows promise as a tool for cardiovascular risk stratification, more robust evidence is required before its integration into routine clinical practice. Future studies should focus on standardizing sLOX-1 measurement protocols and definitions of outcomes.

## Figures and Tables

**Figure 1 biomedicines-13-00444-f001:**
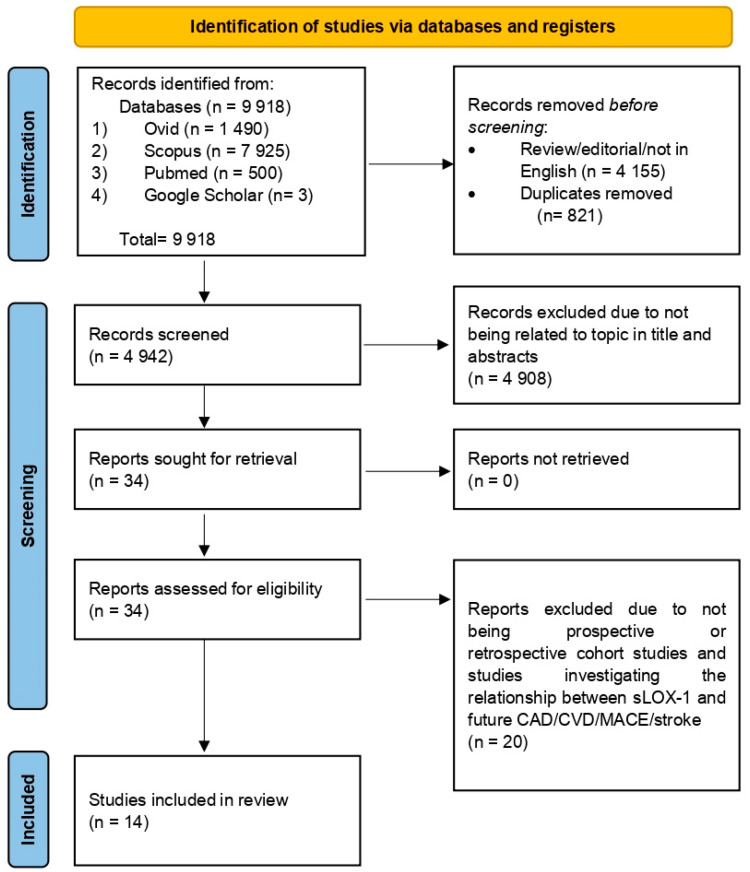
Flow chart illustrating the process of selecting and screening based on PRISMA.

**Figure 2 biomedicines-13-00444-f002:**
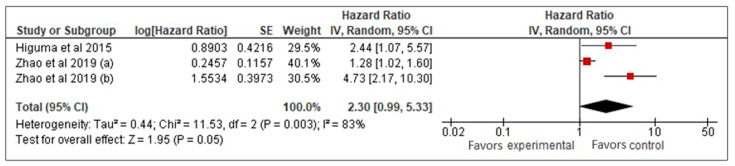
sLOX-1 as a predictor of MACCEs (random-effects model). Red box indicates the effect size for individual study which in this model, the size of the boxes are fairly similar indicating equal weight distribution across all included studies. While, the black diamond, which is the visual representation of the pooled effect sizes, is wide with its right tip touches the line of no effect denoting borderline statistical significance with wide 95% CI.

**Figure 3 biomedicines-13-00444-f003:**
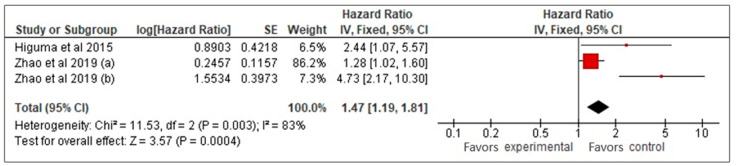
sLOX-1 as a predictor of MACCEs (fixed-effects). Red box indicates the effect size for individual study while the black diamond is the visual representation of the pooled effect sizes. In this model, the most weight (86.2%) is allocated to the study with the biggest red square, thus influencing the pooled effect size (black diamond).

**Table 1 biomedicines-13-00444-t001:** The Newcastle–Ottawa Scale (NOS) for assessing the quality of cohort studies (CAD studies).

Author	Type of Study	Selection				Comparability	Outcome			Total Score
		Representatives of Exposed Cohort	Selection of Non-Exposed Cohort	Ascertainment of Exposure	Demonstration that Outcome of Interest was Not Present at Start of Study	Comparability of Cohorts on the Basis of the Design or Analysis	Assessment of Outcome	Was Follow-Up Long Enough for Outcomes to Occur?	Adequacy of Follow-Up of Cohorts	
		Truly Representative of the Average Community	Drawn from the Same Community as the Exposed Cohort	From Secure Record	Yes	Study Controls for Any Additional Factors	Independent Blind Assessment/Record Linkage	Yes	Complete Follow-Up (All Subjects Accounted for)	
Kraler et al. [24]	Prospective cohort	*	*	*	*	*	*	*		7
Zhao (a) et al. [33]	Prospective	*	*	*	*		*	*		6
Kumar et al. [25]	Prospective	*	*	*	*	*	*	*	*	8
Schiopu et al. [17]	Prospective cohort	*	*	*	*		*	*	*	7
Mashayekhi et al. [18]	Prospective	*	*	*	*		*	*	*	7
Zhao (b) et al. [26]	Prospective	*	*	*	*		*	*	*	7
Higuma et al. [27]	Prospective	*	*	*	*		*	*	*	7

* Indicates one star.

**Table 2 biomedicines-13-00444-t002:** The Newcastle–Ottawa Scale (NOS) for assessing the quality of cohort studies (stroke studies).

Author	Type of Study	Selection				Comparability	Outcome			Total Score
		Representatives of Exposed Cohort	Selection of Non-Exposed Cohort	Ascertainment of Exposure	Demonstration that Outcome of Interest Was Not Present at Start of Study	Comparability of Cohorts on the Basis of the Design or Analysis	Assessment of Outcome	Was Follow-Up Long Enough for Outcomes to Occur?	Adequacy of Follow-Up of Cohorts	
		Truly Representative of the Average Community	Drawn from the Same Community as the Exposed Cohort	From Secure Record	Yes	Study Controls for Any Additional Factors	Independent Blind Assessment/Record Linkage	Yes	Complete Follow-Up (All Subjects Accounted for)	
Zheng et al. [34]	Prospective	*	*	*	*		*	*		6
Ren et al. [28]	Retrospective	*	*	*	*		*	*	*	7
Li et al. [13]	Prospective	*	*	*	*		*	*	*	7
Yan et al. [29]	Prospective	*	*	*	*	*	*	*		7
Yang et al. [30]	Prospective	*	*	*	*		*	*	*	7
Wang et al. [31]	Prospective	*	*	*	*	*	*	*		7
Markstad et al. [32]	Cohort	*	*	*	*		*	*	*	7

* Indicates one star.

**Table 3 biomedicines-13-00444-t003:** Association between circulating sLOX-1 and future cardiovascular events among CAD patients and healthy subjects.

Ref.	Study Design	Population Characteristic	sLOX-1 Measurement	Outcomes Assessed	Key Findings	Conclusion
Kraler et al. [24]	Prospective cohort study. Follow-up at 30 days and 1 year	Patients with ACS (*n* = 2639) ACS = 65.7 ± 1.2 years old	Elisa kit (Thermo Scientific™ Pierce™, Waltham, MA, USA)	Mortality post ACS for all causes and CVD at 30 days and 1 year	(1) ACS patients in the highest sLOX-1 tertile displayed a 2.3-fold increased risk of cardiovascular mortality within 1 year of follow-up (HR, 2.29, 95% CI, 1.19–5.34; *p* = 0.0148). (2) Risk of CVD death at 30 days was increased by 281% in the highest tertile (HR, 3.81, 95% CI, 1.62–19.62; *p* = 0.0036). (3) After multivariable adjustment, high sLOX-1 levels (third tertile) were associated with an increased risk of death from any cause at 30 days (T3: fully adjusted HR, 3.11, 95% CI, 1.44–10.61; *p* = 0.0055). (4) High plasma sLOX-1 was an independent predictor of all-cause mortality over 1 year (HR, 2.04, 95% CI, 1.19–3.92, *p* = 0.0098).	Soluble LOX-1 is a novel and independent biomarker for fatal events in patients presenting with ACS.
Zhao (a) et al. [33]	Prospective study. Review at 2 years	984 patients who were treated for primary PCI; 768 patients had ACS (78.05%); Divided into (1) MACCEs: 69 (59–76) years old, 78.14% men; (2) MACCE-free: 67 (59–74) year old, 70.43% men	ELISA kit (USCN, Wuhan, China).	Composite of MACCEs (all-cause death, readmission for ACS, unplanned repeat revascularization, definite stent thrombosis, and ischemic stroke)	Serum sLOX-1 levels at 2 years were associated with MACCEs (HR 1.278, 95% CI 1.019–1.604, *p* = 0.034).	High baseline serum sLOX-1 concentration predicts 2-year MACCEs and shows an additional prognostic value to conventional risk factors in patients after primary PCI.
Kumar et al. [25]	Prospective study. Follow-up at 1 year	Patients undergoing angiography and diagnosed with ACS and stable CAD. Divided into (1) Group I: patients who underwent coronary angiography (*n* = 18) but did not have established CAD; (2) Group II: patients with stable CAD who underwent percutaneous intervention (*n* = 50); (3) Group III: patients with acute coronary syndrome (*n* = 64); (4) Group IV: healthy controls (*n* = 28)	ELISA kit (USCN, Wuhan, China)	Recurrence of MI and stroke	In recurrence cases (*n* = 9), pre-treatment sLOX-1 level was higher than in non-recurrence cases (*n* = 123); however, the difference was not significant (*p* = 0.655). However, the post-interventional sLOX-1 level was significantly different and higher in recurrence cases (*p* = 0.027).	sLOX-1 is a useful biomarker of stable CAD/ACS and has a potential in the risk prediction of a future recurrence of CAD.
Schiopu et al. [17]	Prospective population-based cohort with 19.5 ± 4.9 years follow-up period	4658 apparently healthy subjects. Grouping following event/no event: (1) No MI: 57.2 ± 5.9; (2) MI: 59.8 ± 5.5; (3) No HF: 57.2 ± 5.9; (4) HF: 61.1 ± 5.0	PEA technique using the Proseek Multiplex CVD96x96 reagents kit (Olink Bioscience, Uppsala, Sweden)	MI or HF	Subjects in the highest tertile of sLOX-1 had an increased risk of myocardial infarction (hazard ratio (95% CI) 1.76 (1.40–2.21) as compared with those in the lowest tertile. No association seen for HF.	There is an association between elevated sLOX-1 and the risk of first-time myocardial infarction.
Mashayekhi et al. [18]	Prospective study. Follow-up at 30 days	320 patients with ACS (236 males and 84 females; mean age 57.29 ± 9.7 years)	ELISA kit (Shanghai Crystal Day Biotech Co. Ltd., Shan hai, China)	In-hospital death, heart failure, and recurrent ischemia	sLOX-1 correlated with MACEs in STEMI and UA/NSTEMI groups (Spearman correlation coefficient = 0.345, *p* < 0.001; Spearman correlation coefficient = 0.189, *p* = 0.017, respectively).	Circulating sLOX-1 could be used as a biomarker to predict major adverse cardiac events in patients with ACS and may be clinically useful in the triage and management of these patients.
Zhao (b) et al. [26]	Prospective study. Follow-up at 2 years	833 patients with stable CAD. Divided into (1) MACE (*n* = 75), aged 68 (39–84) years old, 72% male; (2) No MACE (*n* = 758), aged 64 (32–87) years old, 76% male	ELISA kit (USCN, Wuhan, China)	Composite of MACEs, which were identified as all-cause death, nonfatal AMI, and readmission for Braunwald’s class IIIb UA requiring treatment	Subjects in the highest tertile of sLOX-1 had an increased risk of MACE (HR: 4.73; 95% CI: 2.17–10.30) as compared with those in the lowest tertile.	Baseline sLOX-1 concentrations correlate with 2-year MACEs in stable CAD patients.
Higuma et al. [27]	Prospective study. Follow-up at 3 years	153 patients with STEMI, mean age 67 ± 12 years old, 75% male	Sandwich chemiluminescent enzyme immunoassay (Shionogi Co, Ltd., Osaka, Japan)	All-cause mortality and the combined endpoint of MACEs, which were defined as cardiovascular mortality and recurrent nonfatal MI	Log plasma sLOX-1 level was associated with all-cause mortality (HR: 3.743; 95% CI: 1.853–7.562, *p* < 0.001) and MACEs (HR: 2.436; 95% CI: 1.066–5.566, p = 0.035).	Measurement of plasma sLOX-1 may be useful in identifying patients at high risk for future cardiovascular events.

ACS, acute coronary syndrome; CVD, cardiovascular disease; CAD, coronary artery disease; MI, myocardial infarction; AMI, acute myocardial infarction; HF, heart failure; MACCEs, major adverse cardiac and cerebrovascular events; MACEs, major adverse cardiovascular events; STEMI, ST-elevation myocardial infarction; UA, unstable angina; sLOX-1, soluble lectin-like oxidized low-density lipoprotein receptor-1; HR, hazard ratio; CI, confidence interval; *p*, *p*-value; PCI, percutaneous coronary intervention; ELISA, enzyme-linked immunosorbent assay; PEA, proximity extensive assay.

**Table 4 biomedicines-13-00444-t004:** Association between circulating s-LOX-1 and future recurrent stroke or unfavorable outcomes post stroke among stroke patients and healthy subjects.

Ref.	Study Design	Population Characteristic	sLOX-1 Measurement	Outcomes Assessed	Key Findings	Conclusion
Zheng et al. [34]	Prospective study: follow-up at 3 months	260 patients with AIS. Divided into (1) recurrent ischemic stroke (*n* = 101), age 68.0 (58.0–78.0) years old, men 75.2%; (2) first-ever ischemic stroke (*n* = 165), age 61.0 (55.0–69.0) years old, men 72.7%	ELISA (Solarbio Life Science, Beijing, China)	mRS score at 90 days’ follow-up. An mRS score of 0–2 at follow-up was defined as a favorable outcome and 3–6 as an unfavorable outcome	(1) No correlation was found between sLOX-1 levels and mRS score at 3 months in all patients with AIS or with first-ever stroke. (2) After adjusting for age, admission NIHSS score, NLR, and other variables in the binominal multivariate logistic analysis, sLOX-1 levels remained an independent predictor of unfavorable outcomes in patients with recurrent ischemic stroke with an adjusted OR of 1.489 (95% CI: 1.204–1.842, *p* < 0.0001).	Diagnosis and prognosis are different between patients with recurrent stroke and those experiencing a first-ever stroke. Additionally, sLOX-1 levels serve as an independent prognostic marker in patients with recurrent stroke.
Ren et al. [28]	Retrospective study: review at 12 months	199 patients with AIS and TIA. Divided into (1) Non-recurrence group (*n* = 158): 63.56 ± 12.98 years old, 64.6% male; (2) Recurrence group (*n* = 41): 67.10 ± 9.96 years old, 78% male	Not mentioned	Stroke recurrence based on imaging (*n* = 30) or new neurological deficit symptoms (*n* = 11)	sLOX-1 levels were independent risk factors for stroke recurrence (HR: 1.001, 95% CI: 1.000–1.002, *p* = 0.002).	sLOX-1 levels can be used as a supplement to HR-MR-VWI to predict stroke recurrence.
Li et al. [13]	Prospective study: review at 1 year	272 patients with AIS aged 63.04 ± 8.92 years old, 176 (64.7%) were men 1 year	ELISA	mRS score: favorable or non-favorable outcome. Scores 3 and above considered poor outcome	sLOX-1 was an independent predictor for unfavorable functional outcomes in stroke cases with an adjusted OR of 2.946 (95% CI: 1.788–4.856, *p* < 0.001).	sLOX-1 could be used to predict the long-term functional outcome of stroke.
Yan et al. [29]	Prospective study: review at 3 months	127 ACI patients (first-ever stroke) (78 males and 49 females). Divided into (1) Healthy: 58.42 ± 8.93 years old; (2) No stenosis: 75.4 ± 11.72 years old; (3) Mild stenosis: 62.08 ± 11.25 years old; (4) Moderate stenosis: 61.4 ± 16.04 years old; (5) Severe stenosis: 66.75 ± 13.53 years old	ELISA	mRS for functional recovery, scores 3 and above considered poor outcome	After adjusting for all confounders, sLOX-1 levels were significantly associated with poor functional outcomes, with an OR of 1.005 (95% CI: 1.002–1.007, *p* < 0.001).	The level of sLOX-1 could serve as a useful biomarker to predict the functional outcome of ACI.
Yang et al. [30]	Prospective study: follow-up at 3 months	207 patients with small-artery atherosclerotic occlusion cerebral infarction. Divided by age and male sex (1) Tertile 1, sLOX-1 level < 2.24 pg/mL= 67 (63.50, 75.00) years old, male 67%; (2) Tertile 2, sLOX-1 level 2.24 to 2.89 pg/mL = 66 (58.00, 72.00) years old, male 74%; (3) Tertile 3, sLOX-1 level ≥ 2.90 pg/mL = 68 (59.25, 73.00) years old, male 65%	ELISA (RapidBio Laboratory, CA, USA)	mRS score at 90 days, where 0 to 2 (able to look after own affairs without assistance) was defined as a favorable outcome and 3 to 6 (unable to look after own affairs or death) was defined as poor outcome	Patients in the lowest sLOX-1 tertile had a higher likelihood of achieving a favorable functional outcome at 90 days (OR: 3.47, 95% CI: 1.21–9.96).	In patients with acute atherosclerosis-related ischemic stroke, circulating sLOX-1 levels are correlated with favorable functional outcomes at 90 days.
Wang et al. [31]	Prospective study: review at 3, 6 and 12 months	1200 patients with AIS or TIA. Divided into (1) 400 recurrent cases aged 64.00 (55.00–72.00) years old, 66.5% men; (2) 800 controls (age- and sex-matched) aged 64.00 (55.00–73.00) years old, 66.5% men	Anti-analyzers by using 2 monoclonal antihuman LOX-1 antibodies (TS 92) (ILB International GmbH, Hamburg, Germany)	Recurrent stroke (ischemic or hemorrhagic), ischemic stroke, and combined vascular events (including ischemic stroke, hemorrhagic stroke, myocardial infraction, or vascular death)	(1) Higher sLOX-1 levels were associated with increased odds of recurrent stroke within 3 months and 1 year of follow-up. The adjusted ORs for the highest tertile compared to the lowest tertile of sLOX-1 were 2.10 (95% CI: 1.40–3.16; *p* for trend < 0.0001) and 2.23 (95% CI: 1.61–3.08; *p* for trend < 0.0001), respectively. (2) Similar findings were observed for ischemic stroke, with an adjusted OR of 1.95 (95% CI: 1.28–2.96), and for combined vascular events, with an adjusted OR of 1.95 (95% CI: 1.28–2.96) within 3 months. At 1 year, the adjusted ORs for combined vascular events were 2.30 (95% CI: 1.66–3.19) and 2.31 (95% CI: 1.64–3.24), respectively. (3) Multivariable-adjusted spline regression models revealed J-shaped associations between sLOX-1 levels and the odds of recurrent stroke, ischemic stroke, and combined vascular events within both 3 months and 1 year.	sLOX-1 levels could independently predict recurrent stroke in patients with AIS or TIA.
Markstad et al. [32]	Cohort study: follow-up for 16.5 ± 3.6 years	4703 subjects with no previous history of stroke	Olink Proseek Multiplex kit (Olink Proteomics AB, Uppsala, Sweden)	Recurrent ischemic stroke	(1) Baseline sLOX-1 was associated with recurrent ischemic stroke after 16.5 years of follow-up. Those in the highest tertile had a hazard ratio of 1.75 (95% CI, 1.28–2.39) compared with those in the lowest tertile after adjustment for age and sex. (2) This association remained significant after further adjustment for factors including current smoking, diabetes mellitus, waist circumference, systolic blood pressure, LDL cholesterol, triglycerides, and CRP, with an adjusted odds ratio of 1.41 (CI: 1.01–1.96, *p* < 0.05).	Circulating sLOX-1 levels correlate with carotid plaque inflammation and risk for ischemic stroke.

AIS, acute ischemic stroke; TIA, transient ischemic stroke; ACI, acute cerebral infarction; ELISA, enzyme-linked immunosorbent assay; mRS, modified rankin score; NIHSS, National Institute of Health stroke scale or score; sLOX-1, soluble lectin-like oxidized low-density lipoprotein receptor-1; HR, hazard ratio; OR, odds ratio; CI, confidence interval; *p*, *p*-value; HR-MR-VWI, high-resolution magnetic resonance vessel wall imaging; LDL, low-density lipoprotein; CRP, C-reactive protein.3.2.3. Meta-analysis.

## Data Availability

The original contributions presented in this study are included in the article. Further inquiries can be directed to the corresponding author.

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
