# Peer review of "Prognostic Value of Lectin-like Oxidized Low-Density Lipoprotein Receptor-1 for Future Cardiovascular Disease Risk and Outcome: A Systematic Review and Meta-Analysis"

_biomedicines, 2025, doi:10.3390/biomedicines13020444_

Round 1

Reviewer 1 Report

Comments and Suggestions for Authors

This manuscript investigates the prognostic value of sLOX-1 in cardiovascular and cerebrovascular outcomes through a systematic review and meta-analysis. The topic is clinically significant and has potential implications for cardiovascular risk stratification. The authors adhered to the PRISMA guidelines and provided a detailed review of the literature. However, there are areas that need improvement in terms of methodology clarity, interpretation of results, and presentation of findings.

1. The research question is relevant and timely, considering the growing interest in sLOX-1 as a cardiovascular biomarker. However, the manuscript could benefit from a clearer articulation of its novelty compared to previous reviews in this area. For instance, the clinical utility of sLOX-1, as discussed in the conclusions, lacks specific examples or pathways for its potential integration into clinical practice.

2. The search terms and databases used are appropriate, but the inclusion of gray literature or manual searches for unpublished data is not mentioned. This omission may limit the comprehensiveness of the review.

3. The heterogeneity in the meta-analysis (I² > 80% in some analyses) is substantial, yet the manuscript does not explore this variability in depth. Subgroup or sensitivity analyses should be considered to identify potential sources of heterogeneity (e.g., differences in sLOX-1 assays, population demographics, or outcome definitions).

4. The use of the Newcastle–Ottawa Scale is appropriate, but the scoring should be better linked to the interpretation of study quality and its influence on the results.

5. The results section provides clear summaries of key findings but would benefit from additional explanation of the confidence intervals, especially where statistical significance is borderline (e.g., OR: 1.53, 95% CI: 0.97–2.43).

6. The forest plots are well-constructed but lack annotations to highlight key results or outliers. Including funnel plots or Egger’s tests would strengthen the assessment of publication bias.

7. The authors discuss the potential role of sLOX-1 in cardiovascular risk stratification but do not adequately address the practical challenges in implementing this biomarker in clinical settings.

8. The biological plausibility of sLOX-1’s association with CVD outcomes is discussed well, but the text could benefit from integration with emerging mechanistic studies.

9. The conclusion is balanced but could better emphasize the need for standardization in sLOX-1 measurement protocols and definitions of outcomes in future studies.

Reviewer 2 Report

Comments and Suggestions for Authors

The authors conducted a review and meta-analysis of studies relating blood levels of soluble oxidized low-density lipoprotein receptor-1 (sLOX-1) to cardiovascular disease outcomes, namely major adverse cardiovascular and cerebrovascular events (MACCE), myocardial infarction (MI), heart failure (HF) and ischemic stroke.  In every case, significant predictive value was found for elevated sLOX-1 values.  The practical utility of these findings is questionable, however, since there appears to be no way of relating the individual studies to a general framework.  This can be deduced from an absence of a correlation between multi-study sLOX-1 levels and CVD outcomes.  From the authors' discussion, it appears that there is a lack of correspondence among the measured sLOX-1 values, which is curious, since most of the studies utilized ELISAs.  How can the vendors sell these ELISA kits, if there are no clinical standards or guidelines?  Apparently, each study set a population of measured values that were identified as high or low and outcomes were categorized based on that population.  There needs to be a detailed discussion of these issues.  Perhaps, a table or figure illustrating the variance or discrepancies of the data would be helpful.  Why can't objectively correct sLOX-1 values be determined?  The authors imply that levels of inflammation determine the sLOX-1 values, so the range of the values is very great.  Perhaps normalization of sLOX-1 values using CRP levels would help.  In Table 3, for Kraler et al, ACS = 65.1 ± 1.2 years old, not 65.1 + 1.2.
